# Performance and scaling behavior of bioinformatic applications in virtualization environments to create awareness for the efficient use of compute resources

**Maximilian Hanussek** [1,2], **Felix Bartusch** [1,2], **Jens Krüger** [1] *

**1** Group of Applied Bioinformatics, University of Tübingen, Tübingen, Germany, **2** High Performance and Cloud Computing Group ZDV, University of Tübingen, Tübingen, Germany

* jens.krueger@uni-tuebingen.de

**Data Availability Statement:** All data used to generate the results of this study are publicly available in the GitHub repository with the name BOOTABLE, [https://github.com/MaximilianHanussek/BOOTABLE]. All used tools

## Abstract

The large amount of biological data available in the current times, makes it necessary to use tools and applications based on sophisticated and efficient algorithms, developed in the area of bioinformatics. Further, access to high performance computing resources is necessary, to achieve results in reasonable time. To speed up applications and utilize available compute resources as efficient as possible, software developers make use of parallelization mechanisms, like multithreading. Many of the available tools in bioinformatics offer multithreading capabilities, but more compute power is not always helpful. In this study we investigated the behavior of well-known applications in bioinformatics, regarding their performance in the terms of scaling, different virtual environments and different datasets with our benchmarking tool suite *BOOTABLE*. The tool suite includes the tools BBMap, Bowtie2, BWA, Velvet, IDBA, SPAdes, Clustal Omega, MAFFT, SINA and GROMACS. In addition we added an application using the machine learning framework TensorFlow. Machine learning is not directly part of bioinformatics but applied to many biological problems, especially in the context of medical images (X-ray photographs). The mentioned tools have been analyzed in two different virtual environments, a virtual machine environment based on the OpenStack cloud software and in a Docker environment. The gained performance values were compared to a bare-metal setup and among each other. The study reveals, that the used virtual environments produce an overhead in the range of seven to twenty-five percent compared to the bare-metal environment. The scaling measurements showed, that some of the analyzed tools do not benefit from using larger amounts of computing resources, whereas others showed an almost linear scaling behavior. The findings of this study have been generalized as far as possible and should help users to find the best amount of resources for their analysis. Further, the results provide valuable information for resource providers to handle their resources as efficiently as possible and raise the user community's awareness of the efficient usage of computing resources.

and datasets are publicly available at a webserver as tar archive [https://s3.denbi.uni-tuebingen.de/max/BOOTABLE.tar].

**Funding:** This work has been supported by the High Performance and Cloud Computing Group at the Zentrum für Datenverarbeitung of the University of Tübingen providing the overall environment including the position of JK, the state of Baden-Württemberg through bwHPC funding the position of FB and the German Research Foundation (DFG)[https://www.dfg.de] through grant no INST 37/935-1 FUGG funding compute resources. Part of the work presented here was also supported through the BMBF funded project de.NBI (031 A 534A)[https://www.bmbf.de] supplying the cloud computing hardware and the position of MH. The funders had no role in study design, data collection and analysis, decision to publish, or preparation of the manuscript.

**Competing interests:** The authors have declared that no competing interests exist.

## Author summary

The analysis of biological data increasingly makes more and more use of computational resources, or would not be possible at all without them. Besides classical high performance computing resources like computer clusters, the technology of cloud computing and its applications in biology and bioinformatics has strongly increased over the last years. With cloud computing, virtualization technologies are also increasingly used. However, computing resources are not endlessly available and should therefore be used as efficient as possible. To support the efficient development of multithreaded applications, we developed our benchmarking tool suite *BOOTABLE* and used it to study the scaling behavior of different bioinformatics tools, covering a wide range of application areas using different computing environments (virtualized and bare-metal). Our study showed that not every tool benefits from higher numbers of CPU cores, also linear scaling properties are not seen for all of them. With this study we want to create an awareness for the responsible usage of computing resources. It is not always better and faster to use more and more resources. Sometimes it is helpful to check, whether a tool or application is capable of using larger resources or not.

This is a *PLOS Computational Biology* Benchmarking paper.

## Introduction

Today's sequencing technologies are becoming more and more sophisticated and produce larger and larger amounts of data on the scale of tera- and petabytes in mostly every -omics area (genomics, proteomics, metabolomics). In order to analyze such huge amounts of data on a large scale, advanced algorithms and applications, developed by bioinformaticians, are becoming more and more important to answer the underlying biological questions. Smart algorithms and their efficient implementation are one part. The other part are the required resources to analyze large amounts of data. Algorithms in bioinformatics mostly try to solve NP-Hard problems [1]. This problem type shows an exponential increase in its runtime with the increase of the input size. To solve such a problem or to find sufficient approximations, large computing resources are required. Apart from the area of sequencing analysis there are many more topics that would also benefit from larger compute resources, like molecular docking, molecular dynamic simulation or, currently one of the most interesting topics, machine learning. Almost everywhere an easy access to high performance computing resources is required. Access to such resources can be gained by commercial computing cloud providers like Amazon, Google or Microsoft [2–4]. Academic cloud providers are for example the de.NBI cloud [5], bwCloud [6], or a hybrid one like Helix Nebula [7]. No matter which of the providers or even an own solution is preferred, compute resources are valuable and these resources are usually shared with others. Therefore, an efficient utilization is necessary to minimize the spent time and costs. To use the offered resources efficiently, applications must be efficient as well. Some applications can benefit from multiple CPU cores due to their underlying algorithms or implementation, others not. Therefore, it would be desirable for users and resource providers to know in advance, how many resources, like CPU cores, memory and storage are reasonable to conduct computations most efficiently. Nowadays only supercomputer centers request smaller scaling studies in beforehand to see whether a given application performs as expected and makes use of the applied resources. The hereby addressed scalability is one factor, another factor are the more and more used virtualization technologies in particular due to the increasing offers of compute

clouds. Such compute clouds are usually providing access to virtual machines but not directly to the hardware, like for high performance computing (HPC) clusters. What kind of effect could that have on the used tools and applications?

In this study, we focus on the issues of scaling, the impact of different virtualization environments and datasets for widely used bioinformatic applications. First, the paper gives an introduction about the term scalability and virtualization technologies, followed by the Related Work section and descriptions how the measurements have been conducted, including brief explanations of the used applications and datasets. The main part describes the gathered results, how these results can be explained and generalized. The work finishes with a conclusion and an outlook on future work.

## Scalability

Scalability in general can be defined as a term that describes the ability of a system to process a growing amount of work, objects or elements in a decreasing or stable amount of time by adding additional resources to an initial system [8]. A more specific part is software scalability. Software scalability is an important topic for bioinformatics applications and software as the underlying algorithms mostly handle problems with a quadratic time complexity or even worse often belonging to the class of NP-Complete or also NP-Hard problems (non-deterministic polynomial-time) [9]. These problems are present in mostly every area belonging to bioinformatics, like multiple sequence alignments, protein folding predictions or phylogenetic reconstructions [10]. Due to this, most software developers tried to make use of different parallelization strategies in order to reduce the execution time, like using multiple cores and processors on a single machine with shared memory from the very beginning. Another approach is to use multiple compute nodes connected to a compute cluster with distributed memory. Also modern programming models and implementations like the MapReduce approach, introduced by Google [11], or Apache Spark [12] have been invented especially for the topic of big data analyses to make computations feasible in reasonable time [13].

From all these interesting strategies above we put our focus on the parallelization strategy using multiple cores of a single compute node with shared memory, also known as multithreading. The concept of multithreading as programming and execution model can be explained as follows, a single process is started, that acts as a parent process and further is able to spawn threads. These threads can be executed independently of other threads which makes parallelization possible. On systems with a single core, threads would be handled by slicing and distributing the available CPU time over all running threads. On more modern compute systems with multi-core architectures it is possible to distribute running threads over the existing CPU cores so that each thread is dedicated to its own CPU core which makes it unnecessary to slice and share the available CPU time. Support for multithreading is part of most available programming languages (Java, Python, C, C++), providing an API (application programming interface) to standardized thread interface implementations. Widely used APIs are POSIX Threads (Pthreads) [14] and Open Multi-Processing (OpenMP) [15]. These have already been compared with each other regarding their performance. The selected use-case was the prediction of protein secondary structures [16]. Most of the bioinformatics tools in this scaling study are using either Pthreads or OpenMP to implement multithreading, an overview is stated in Table 1.

## Virtualization technologies

Virtualization can be divided into two major concepts, hypervisor-based virtualization and container-based virtualization [17]. Both approaches can be illustrated as different layers,

**Table 1.** *BOOTABLE* **benchmark applications, including their general category, version, used multithreading API and the GCC compiler version, whereas an already compiled binary is used for SPAdes BBMap and SINA.** TensorFlow is installed via Python pip. For BBMap, implemented in the programming language Java, no note about the used multithreading API could be found.

| Application | Category | Version | Multithreading API | GCC |
|---|---|---|---|---|
| BBMap | Sequence Alignment | 38.87 | - | - |
| Bowtie2 | Sequence Alignment | 2.4.2 | Threading Building Blocks | 4.8.5 |
| BWA | Sequence Alignment | 0.7.17 | Pthreads | 4.8.5 |
| Velvet | Sequence Assembly | 1.2.10 | OpenMP | 7.3.0 |
| IDBA-UD | Sequence Assembly | 1.0.9 | OpenMP/Pthreads | 4.8.5 |
| SPAdes | Sequence Assembly | 3.12.0 | Pthreads | - |
| Clustal Omega | Multiple Sequence Alignment | 1.2.4 | OpenMP | 4.8.5 |
| MAFFT | Multiple Sequence Alignment | 7.475 | Pthreads | 4.8.5 |
| SINA | Multiple Sequence Alignment | 1.7.2 | Threading Building Blocks | - |
| GROMACS | Molecular Dynamics | 2018.3 | OpenMP | 7.3.0 |
| CIFAR-10 model build | Machine Learning | 1.4.0 | Python Threadpool | - |

starting from the hardware layer at the bottom to the application layer at the top. A comparison is illustrated in Fig 1.

The following subsections explain both technologies in more detail and show well-known representatives.

**Hypervisor-based virtualization.** Well-known examples for hypervisor-based virtualization software are VMware [18], Xen [19] or KVM (Kernel-based Virtual Machine) [20]. KVM is the most used option in open source environments. It is widely used, integrated directly into the upstream Linux Kernel and together with the libvirt virtualization platform the standard combination used by the cloud operating software OpenStack [21, 22]. Further, KVM supports hardware assisted virtualization requiring a processor equipped with hardware extensions like Intel VT-x [23] or AMD-V [24]. One general aspect of hypervisor-based virtual machines is the creation of a full virtual machine with its own guest operating system (guest OS) and kernel. Whereby the hypervisor layer, sitting between the guest OS and the host operating system (host OS), translates the instructions and thus enables virtualization.

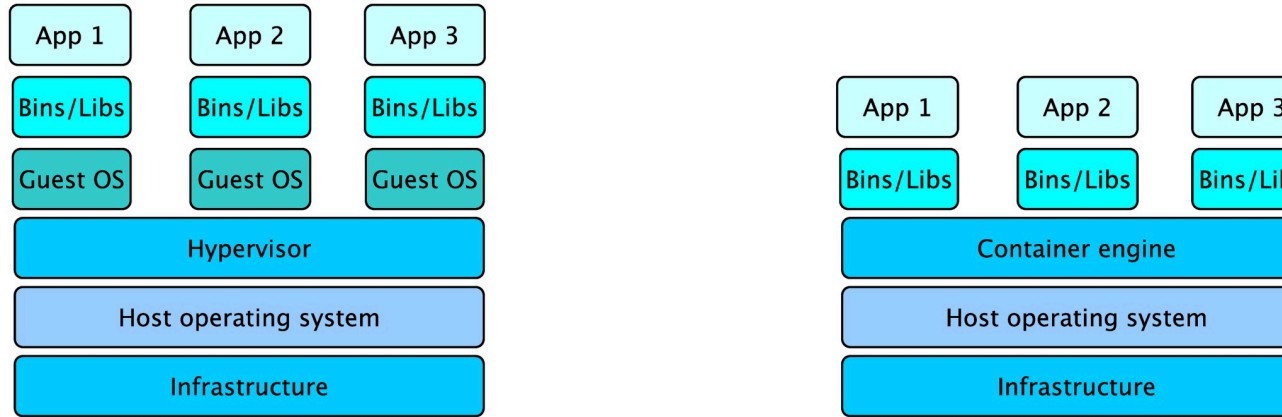

**Fig 1. Comparison of hypervisor-based and container-based virtualization.** Schematic illustration of hypervisor- and container-based virtualization. The container technology concept with the container engine on top of the host operating system is shown on the right side, whereas the hypervisor technology is illustrated on the left side.

**Container technologies.** In contrast to a hypervisor-based virtualization approach, a container based approach is a more lightweight virtualization [17]. The most well-known representatives nowadays are Docker [25] and Singularity [26]. The lightness of container virtualization comes through the usage of the host OS without using an intermediate hypervisor layer and a full standalone guest OS and no own kernel. Without these limiting factors it is possible to start a larger number of applications compared to a full virtual machine as less resources are needed. But saving the hypervisor layer and using the host OS itself has some drawbacks. The operating system running in the container needs to fit to the host operating system. There are some exceptions regarding Docker containers with a Linux based operating system on a windows host system but this is not further elaborated here [27]. Another problem of containers are security concerns. As the approach of container virtualization uses the host OS more or less directly, it can be accessed through the container, which can result in permission conflicts or privilege escalations inside a container. For example, inside a Docker container, a user can get root permissions, but if a directory of he host system is mounted inside the container, which should not be accessible, it can be edited or deleted because inside the container a user has the correct permissions [28]. Over time mechanisms have been implemented to prohibit such a behavior but still the docker daemon is running as root process and can always be a potential risk. Singularity on the contrary is especially designed to run for example in HPC environments where users usually get no privileged access at all. The difference to Docker is, that Singularity does not require a daemon process with root privileges. A singularity container runs with the same user id as on the host system and therefore has the same permissions, inside and outside of a container. Furthermore, containers are a lightweight solution to conserve specialized runtime environments, that can be used to distribute software more easily or to increase the reproducibility of scientific workflows [29].

## Related work

Due to the increase of available HPC resources, especially in the area of bioinformatics, the scalability of tools gets more and more important. Throughout our literature search we have found many papers covering qualitative aspects, regarding the results of various tools, mostly settled in the area of read assembly and sequence alignment tools [30–32]. Even through an exhaustive search in the literature we have not been able to find many comparative studies focusing solely or even in parts on the scaling behavior of multithreaded applications in bioinformatics. The number of papers discussing practical use-cases is even lower as many papers are discussing possible scaling strategies from a more theoretical point of view.

An example of a more theoretical paper is from Yang et al. [13], considering different approaches on the handling of big data analysis in reasonable time. Yang et al. give an introduction about the terms scalability, also providing a section about different computing infrastructures, like cluster computing, grid computing and cloud computing, decreasing the required runtimes by adding more resources or using special hardware like GPUs. Yang et al. present different approaches using the MapReduce or Apache Spark programming models in order to handle big data analysis more efficiently.

A more practical focused work is the paper of Janga and Cherukuri [33]. Cherukuri et al. have compared different *de novo* assembly tools with regards to the optimal performance of OLC (Overlap Layout Consensus) approaches on Nanopore data. This work compares the assemblers Velvet, ABySS, Celera and SSAKE regarding different quality measures like the number of resulting contigs, N50 values or the total length of the contigs by varying the number of reads. The full input datasets are split into chunks of 25%, 50%, and 75% of the full dataset to measure the impact of the number of reads on the mentioned assemblers. This study

also contains measurements about wall-time and memory usage, also related to the varying number of input reads which is interesting in order to estimate runtimes and memory requirements for further usage. But unfortunately it is not stated on what kind of hardware the performance measurements have been conducted.

The work of Lenis and Senar [34] is one of the few having a view on a variety of sequence alignment tools regarding their scaling behavior on multithreading, using different memory allocation strategies. Their paper explains, how modern compute architectures look like and how different memory allocations can influence the performance of tools making use of NUMA (Non-uniform memory access) architectures. Furthermore, Lenis et al. compared different memory allocation strategies like local allocation and interleaving but also two different hardware manufacturers (Intel, AMD) and also the number of threads made available to the different alignment tools. The covered applications also include Bowtie2, that is also used in our study.

As we could not find any more comparative studies regarding the scaling behavior of multithreaded tools in the literature we broadened the search to find at least studies of the involved benchmarking tools itself. The extended search revealed some more information of already published work, covering the same aspects of our work but not for every application in the benchmark set.

For the machine learning framework TensorFlow and its applications, lots of papers are available, as machine learning is a very hot topic in these days. One example is the paper from Ramirez-Gargallo et al. [35]. The paper describes how state-of-the art HPC clusters and their configuration in hardware (different processor architectures), but also in software (arithmetical libraries), can affect the scaling and runtime behavior. One part of the work addresses the intra-node evaluation. In this part Ramirez-Gargallo et al. compared the scaling behavior of a TensorFlow image recognition workflow on three different test clusters (MareNostrum4, Power9, Dibona), but only on a single node of the whole cluster. The three different clusters are also reflecting different CPU architectures, MareNostrum4 is based on Intel Xeon processors, whereas Power9 uses IBM Power9 processors and the Dibona cluster runs ARM processors. The first part discusses the thread behavior of the different cluster nodes using the AlexNet model. The thread sizes are chosen as power of two, starting with one thread up to 64 threads. Further variations has been made regarding the batch sizes, depending on the used type of cluster node and the used arithmetic libraries like MKL for the x86 architecture and the ARM performance library. In the further sections, Ramirez-Gargallo et al. transferred the best found combination of hardware and software of the intra-node results to multiple nodes, which will not be described in detail here. Different to our study is that other CNN (Convolutional Neural Network) training-models, has been used (AlexNet and ResNet50). A further difference to Ramirez-Gargallo et al. is, that our work includes unusual thread sizes in increments of 1, in the range from 1 to 36 to smooth the curve of the scaling behavior and to consider two socket server systems, where each CPU comes with 14 cores for example and therefore partitions of seven cores could make sense.

Another tool that has a sufficient amount of scaling publications is the molecular dynamics software GROMACS (Groningen Machine for Chemical Simulations). In the work of Hess et al. [36] three different datasets, varying in their particle sizes, are used with GROMACS on the SuperMUC. The goal of the paper was to find the best variation of node numbers and thread numbers, achieving the highest performance with respect to the simulated nanoseconds per wall-clock time. The whole benchmark setup is described in detail, including the datasets and the reason of their choice. The paper is strongly focused on inter-node parallelization than intra-node parallelization.

For both assembly tools considered, SPAdes and IDBA, we were not able to find a scaling study that we could compare this work to. For IDBA not a single study could be found through our literature search, regarding wall-times or thread numbers. Similar results have been observed for SPAdes. Most of the studies, like the original paper [37], are investigating the quality of the results but not the computational efficiency. The same holds for Clustal Omega, where only papers have been found that mention the predecessor, ClustalW [38] or Clustal Omega in comparison with T-Coffee [39] but without varying the thread numbers also for Clustal Omega.

Another factor that can also affect scalability is the use of different virtualization methods. Throughout the literature search, papers have been collected that evaluated the different performances between full virtualization, para-virtualization and container technologies [40–42] in various combinations. Most of them performed a comparative analysis on a technical level like the work of Arango et al. [41]. It compared different containerization technologies with each other, also related to the bare-metal performances, focused on parameters like CPU performance, measured with HPL (High-Performance Linpack), network throughput, disk read and write performances and also memory usage measurements. A quite new publication of Radchenko et al. [42] from 2019 is going a step further. In their study they also compared technical performance parameters between fully virtualized environments (KVM, Xen) and Docker, but also include specific benchmark applications from machine learning (K-Means, Logistic Regression), graphs processing problems like Single-Source Shortest Paths and SQL queries.

The most similar work to the here presented study that has been found, is from Estrada et al. [22]. Estrada et al. compared different virtualization technologies to each other also including containerization with LXC containers. In contrast to the works mentioned before, the performance measurement is done on application level. The sequence aligners BWA and Novoalign are used as benchmark applications to cover real world examples. The benchmarks are conducted with CPU pinning enabled and disabled as certain tools could benefit from it. This work gives an insight how virtualization in different ways can have effects on real world applications especially in the field of bioinformatics.

## Methods

As mentioned in the Introduction section, we used our own benchmark tool suite *BOOTABLE* [43] to measure the scaling behavior of different bioinformatics applications in a well-defined software environment. The tools and datasets contained in *BOOTABLE*, are described in more detail below.

### System environment

In total we used three different kinds of computational environments, but all are using the same underlying hardware. We conducted every benchmark directly on a server (bare-metal), in a virtual machine (VM) embedded in an OpenStack cloud, that is part of the de.NBI Cloud [5], and the dockerized version of *BOOTABLE* inside a VM. The OpenStack cloud software is used to launch and manage the virtual machine, but has no direct impact on the VM itself, every other virtualization management software should be just as good. Docker has been chosen as it is one of the mostly used container technologies especially in the community of bioinformatics. The hardware consisted of a two socket server system equipped with Intel Xeon Gold 6140 "Skylake" CPUs with 18 cores each, which adds up to 36 CPU cores in total, operating at a base clock rate of 2.3 GHz. Hyper-Threading has been disabled and the ratio of physical CPU cores to virtual CPU cores was 1:1, which means, every virtual core has been mapped

to exactly one physical core. Further, the system hardware provides 1.5 TB of RAM and a local SSD with a capacity of 480 GB. For all three environments, CentOS in version 7 has been used as operating system. The virtual machine environment is using QEMU-KVM as virtualization software, whereas Docker has been used in version 1.13.1 and overlay2 as storage driver. The dockerized version of *BOOTABLE* has been created from a Dockerfile, that is similar to the *BOOTABLE* installation script. From the Dockerfile a Docker image has been build on the same server type as the benchmarks has been performed on. For the scaling benchmark, the image has been pulled from Docker Hub and run as an interactive container, without any further adjustments.

## BOOTABLE

Our benchmark tool suite *BOOTABLE* has been initially developed to let resource providers explore and evaluate their hardware with regards to bioinformatic workloads. But we did not stop there and extended *BOOTABLE* to make it a valuable tool for application developers too. With the rich output, developers can get an overview or detailed insights about a tool's resource consumption and scaling behavior, we put the focus in this work. *BOOTABLE* comes with lots of features, like an easy installation process covering multiple environments. It can be installed via the provided installation script, that takes care of the required tools and datasets as well as the provided Ansible [44] playbook. Furthermore, *BOOTABLE* is available as a Docker and Singularity image, but also as a virtual machine image in the QCOW2 format. The benchmark execution is kept simple but still offers enough configuration parameters. Users can choose between different datasets varying in size and complexity, diverse sets of tools covering general parts of bioinformatics and beyond. It is further possible to request the number of threads, that should be used by the chosen tools, just as the number of replicas that indicates how many times the same benchmark is repeated to get more accurate results, reducing the impact of outliers as some of the tools make use of random numbers as starting point, for example. An additional feature we focused on in this paper is the scaling mode. If the scaling option is selected, the desired tools are executed using one core, a quarter, half and all available CPU cores. We modified the regular execution script, to capture all possible numbers of threads starting from one to the maximum, increasing in steps by one. The modified script is also available under (https://github.com/MaximilianHanussek/BOOTABLE/tree/master/mods). After a benchmark run is finished, different measurements with different metrics are available. These are subsequently used to create a more human readable report. If scaling benchmarks have been performed, specific graphs are created to illustrate the behavior of the chosen tools. In this study the benchmarks were performed by *BOOTABLE*, using five replicas for each CPU core run, all available predefined datasets (short, middle, long), and all available tools. The used tools are explained in more detail in the upcoming Tools section and Tables 1 and 2. The same applies to the available datasets, which are described in-depth in the Datasets section and Table 3.

## Tools

The applications included in *BOOTABLE* cover diverse application fields of bioinformatics or life sciences, like sequence assembly, (multiple) sequence alignment, molecular dynamics and machine learning. A complete enumeration is represented by Tables 1 and 2, with Table 2 focusing on the algorithms.

**Tool selection.** The tools were selected based on different criteria. First of all, a tool has to make use of multithreading in most parts of their implementation, to perform a scaling study. Further, a tool should be well documented, both in terms of its execution, implementation and

**Table 2. List of used benchmark tools.** The algorithm column describes in brief the used algorithms to allow a comparison among each other (* No parallelization).

| Application | Algorithm |
|---|---|
| BBMap | Index: Sliding window with k-mers, Align: k-mer look up and scoring |
| Bowtie2 | Index: BWT and FM-Index, Align: Seed and extend using Dynamic Programming |
| BWA | Index: BWT and FM-Index*, Align: Seed and extend using Dynamic Programming |
| Velvet | De Bruijn Graph based on k-mers |
| IDBA-UD | Iterative de Bruijn Graph based on increasing k-mers |
| SPAdes | Multi-sized de Bruijn Graph, kmers for graph construction only, graph theoretical operations |
| Clustal Omega | Pairwise distance matrix, guide tree (k-means), profile alignments (Hidden-Markov Models) |
| MAFFT | Pairwise distance matrix, guide tree (UPGMA), group-to-group alignment (iteratively) |
| SINA | Index: Reference sequence selection used to create partial order MSA, Align: modified Needleman-Wunsch |
| GROMACS | Integration of Newton's equation of motions, Force field (bonded and non-bonded interactions, domain decomposition) |
| CIFAR-10 model build | Convolutional neural network, multilayer architecture, alternating convolutions, softmax classifier |

algorithm. In addition, the selected tools should have as less dependencies as possible, like required tools or datasets. Of course it should be relatively popular and free for academic use, that many users are able to profit from the findings of this study. Throughout the literature search and communications to users of the de.NBI Cloud, the tools explained in the following subsections have been selected. Unfortunately, other popular tools could not be considered, because they did not meet certain criteria. For example the multiple sequence alignment tool MUSCLE [45] does not support multithreading. T-COFFEE [46] could not be considered either, since access to databases is necessary. We make no claim to completeness regarding the selected tools. However, to the best of our knowledge, the selected tools should cover a large range of application areas and users.

**Table 3.** *BOOTABLE* **benchmark datasets with sizes in Megabytes and the number of sequences or images for CIFAR-10 or number of atoms for ADH in the Items column.**

| Dataset | Size | Items |
|---|---|---|
| ERR2510006.filt.fastq | 1600 | 6,478,764 |
| ERR016155.filt.fastq | 284 | 1,185,766 |
| ERR015528.filt.fastq | 226 | 882,933 |
| SRR741411.filt.fastq | 136 | 526,501 |
| DRR001012.fastq | 683 | 909,677 |
| DRR001025.fastq | 2000 | 13,472,829 |
| GRCh38_full_analysis_set_plus_decoy_hla.fa | 3100 | 3366 |
| wgs.ANCA.1.fsa_aa | 0.5 | 1258 |
| SILVA_138.1_SSURef_NR99_12_06_20_opt.arb | 541 | 510,984 |
| GTDB_bac-arc_ssu_r86.fa | 31 | 21,559 |
| RefSeq-RDP16S_v2_May2018.fa | 23 | 15,336 |
| OE-38_R1.fastq | 3.7 | 6613 |
| CIFAR-10 | 163 | 60,000 |
| ADH_bench_systems (adh_cubic) | 59 | $\tilde{1}34,000$ |

In the category of sequence assembly tools, Velvet, IDBA and SPAdes were selected as all of them fulfill the criteria of multithreading, are well documented and straightforward to install. In addition all are using the general concept of de Bruijn graphs but in different variants, which might be of interest to compare them with each other. Velvet was chosen because of the generated workload, as a large part of the implementation is single threaded but some parts are multithreaded. IDBA is selected as no article was available examining the scaling performance, the same applies for SPAdes. Further, SPAdes was chosen because it uses larger amounts of RAM, which is interesting to observe effects of larger data sets and RAM usage on the scaling behavior.

The category of sequence alignment is covered by the tools BBMap, Bowtie2 and BWA, All are well-known, especially Bowtie2 and BWA. Both are using similar approaches for their index structures and for the alignment step. This makes it interesting to compare them with each other and with BBMap, which uses a different approach.

The tools Clustal Omega, MAFFT and SINA were chosen as representatives for multiple sequence alignment tools. Especially Clustal Omega and MAFFT have a large user base. The implemented approach of both tools is similar, therefore a comparison might be interesting. With SINA, a tool specialized on rRNA sequences, is selected to cover this specific topic.

For the topic of molecular dynamics, we decided to choose GROMACS. GROMACS is widely used, freely available, very well documented and we are very familiar with it. Of course there are other tools available, like CHARMM (Chemistry at HARvard Macromolecular Mechanics) [47] or Amber [48], which also have a large community and would have also been a valid choice.

For the area of AI, specifically machine learning, the TensorFlow framework and the CIFAR-10 application have been chosen as TensorFlow is one of the mostly used frameworks and the CIFAR-10 application image recognition can be applied in the context of medical images. Of course there exist other widely used frameworks like PyTorch [49], for example.

A brief description of the tools and their application area is shown in the following.

**Sequence assembly.**  Sequence assembly is a method to reconstruct genomes from reads, generated by sequencers. Genome or sequence assembly tools take advantage of the fact that gathered reads are overlapping with each other. In the first step small reads are assembled to larger fragments, known as contigs. Furthermore, contigs can be linked to each other in order to build scaffolds. In the end, all parts can be merged to reconstruct the whole input sequence [50].

**Velvet**. Velvet [51] is a *de novo* assembler using a de Bruijn Graph structure to assemble given reads based on k-mers.

Taking a look on the time and memory complexity, the main bottleneck of Velvet is the graph construction step, that depends on the input size, the number of reads and the chosen *k*, as these parameters define the number of nodes. Another factor is the complexity of the underlying data. More complex data leads to a more convoluted graph, that requires a longer runtime. The complexity consists mainly of the parameters coverage, error rate and number of repeats in a sample.

**IDBA-UD**. IDBA-UD is another widespread assembly tool, developed for the application on metagenomic and single-cell sequencing data showing a highly uneven depth [52], based on a de Bruijn Graph data structure using paired end reads and an iterative approach based on different k-mer lengths.

Regarding runtime and complexity, the IDBA-UD algorithm is similar to Velvet. The larger the input size and the complexity of the input data the larger and complex is the build graph and therefore the higher the runtimes.

**SPAdes**. SPAdes (St. Petersburg genome assembler), developed by Bankevich et al. [37], is another tool belonging to the category of *de novo* assembly tools. SPAdes also uses the concept of a de Bruijn Graph structure and k-mers. But the k-mers are only used to construct the multi-sized de Bruijn Graph. Afterwards, generic graph theoretical operations are performed. In the end a condensed edge representation of a paired assembly graph [53] is constructed and the reads are mapped back to the created contigs.

Bankevich et al. conducted a performance benchmark using the error corrected ECOLI-SC read set. The results show that the first two steps of the algorithm take most of the time. Especially the first one handling the graph construction uses most of the total execution time, as SPAdes iterates over a list of different *k*s, with each iteration taking nearly the same amount of time. In general it applies, the higher the number of reads as input data, the higher the number of nodes the graph consists of and the longer the runtime. Other parameters that have an impact on the execution time are the tangledness of the graph, which can be reduced by performing a pre-error correction, for example using BayesHammer [54].

**Sequence alignment.** Sequence alignment can be defined as a procedure where a reference sequence is used and other sequences are matched (aligned) against the reference sequence. This method makes it possible to find similarities in specific regions of the reference sequence, that can be used further to find functional or evolutionary correlations, for example [55].

**BBMap**. BBMap (Bestus Bioinformaticus Map) [56] is a splice aware alignment tool designed for small and long reads resulting from DNA and RNA sequencing. BBMap uses a sliding window approach based on k-mers to create the index structure. The alignment is based on using triplets, holding the information of possible hits [57].

The space and time complexity dependents on the number of bases of the reference sequence and also on the chosen k-mer size. Important for the alignment step is the number of reads and their number of bases. Also the k-mer size is of importance. The smaller the k-mers the more sensitive is the mapping but the more amount of time and space is required.

**Bowtie2**. Bowtie2 is the successor of the widely used read alignment tool Bowtie [58]. The tool is named after the underlying implementation of the Burrows-Wheeler-transform [59]. Bowtie2 uses the Full-text Minute-size index (FM-Index) [60], to handle even large amounts of sequencing reads as fast as possible. Bowtie2 uses Burrows-Wheeler ranges to find promising candidates to start a dynamic programming alignment approach from. In the end a Sequence Alignment Map (SAM) is constructed.

The runtimes of the different steps reveal, that the initial step creating the index structure, is the computationally most expensive one. But once the index structure is computed it can be reused for subsequent alignments. Due to the advantageous data structure of the FM-Index, the alignment can be performed quickly. The rate-determining variable is again the size of the input data, for the index construction the number of bases of the reference genome, for the alignment step the number of reads. The final alignments, using a dynamic programming approach, result in a time complexity of $\mathcal{O}(NM)$ where $N$ is the length of the reference sequence and $M$ the length of the sequence to be aligned.

**BWA**. BWA (Burrows Wheeler Aligner) is a software package consisting of different alignment algorithms [61]. This work focuses on the BWA-MEM (Burrows Wheeler Aligner—Maximal Exact Matches) algorithm as it is the preferred one when using BWA [62]. As Bowtie2, BWA is using the FM-Index for the index structure of the reference sequence. The initial seed alignment is done by searching for SMEMs (Supermaximal Exact Matches) [63], followed by a affine-gap-penalty dynamic programming approach for the alignment extension. The general time and space complexity can be found in the original publication [61].

**Multiple sequence alignment (MSA).** A multiple sequence alignment deals with a similar problem as a pairwise sequence alignment but instead of comparing only two sequences with each other, multiple sequences (three or more) are compared with each other. The sequences to be compared can consist of nucleotides or amino acids. Results of such alignments can be used to find homologies between samples for subsequent phylogenetic analysis. [64].

**Clustal Omega**. Clustal Omega [65, 66] is a tool made for the alignment of multiple sequences in a large number. Clustal Omega uses a guide-tree approach (mBed algorithm), in combination with the bisecting k-means algorithm [67] to cluster pairwise sequence distances. Afterwards, profile-profile alignments are constructed, making use of HHalign [68].

The runtime complexity and memory usage shows that Clustal Omega reduces the runtime and memory in multiple ways. The mBed algorithm is used for the pairwise distance calculations, that reduces the complexity from $\mathcal{O}(N^2)$ where $N$ is the number of sequences, to $\mathcal{O}(N(\log(N))^2)$. The reduction of complexity is achieved by using not all but only up to $(\log N)^2$ sequences of all available ones. Further is the memory footprint rather low, avoiding a full distance matrix. Important for the complexity observations in the second part is the profile-profile alignment step. It can be represented as a function dependent on $N$, the number of sequences, the profile length $L$ and the shape of the guide tree. A usual MSA needs $N - 1$ profile-profile alignments. Therefore, the number of alignments increases linearly with the number of sequences as well as the runtime. An increase of the sequence length would also increase the length of the profile sequences as HMM (Hidden Markov Model) matrices require a multiple of $L_1 \times L_2$, leading to a quadratic increment in the time complexity. The shape of the guide-tree contributes in that way to the overall runtime, that a balanced tree can be traversed faster than an imbalanced, chained one. Further, the runtime is reduced by implementing multi-threading for the pairwise distance calculations and the alignment match states computations, using the OpenMP library [69].

The space complexity of a profile-profile alignment with length $L$ can be expressed by the term $8 \times 6 \times L_1 \times L_2$, where 8 is the space required for a double variable in Bytes and 6 the number of matrices, created by HHalign.

**MAFFT**. MAFFT is a multiple sequence alignment program using the fast Fourier transform (FFT) method. The software was initially published in 2002 [70] and has been continuously developed since then. MAFFT uses the concept of a guide tree (UPGMA) to create an initial MSA by group-to-group alignments, that are refined in a subsequent iterative procedure.

The time complexity of MAFFT, using the progressive method, is expressed by the length ($L$) and number ($N$) of the input sequences with the term $\mathcal{O}(N^2 L)\mathcal{O}(NL^2)$. The first part of the term is related to the guide tree construction and the second part to the group-to-group alignment. The space complexity is specified with $\mathcal{O}(N^2) + \mathcal{O}(L^2) + \mathcal{O}(NL)$ [71–74].

**SINA**. SILVA Incremental Aligner (SINA), developed by Glöckner et al. [75], is a multiple sequence alignment tool, specialized on ribosomal RNA (rRNA), using the rRNA databases provided by the SILVA project [76]. The algorithm of SINA uses a mix of partial order alignment [77] and k-mer sequence search [78].

Due to the sequence selection step, prior to the alignment step, time and space complexity are decoupled from the size of the reference MSA.

**Molecular dynamics.** Molecular dynamics deals with the simulation of atom and molecule movement and therefore their physical behavior. One use case in bioinformatics, for example, is protein structure prediction.

**GROMACS**. GROMACS, the Groningen Machine for Chemical Simulations [79] is a very popular molecular dynamics tool suite in the area of open source applications. GROMACS is

mostly used for the simulation of bio-molecule dynamics, based on Newton's equations of motion. Within a periodic molecular system in theory all particles are interacting with each other, but in practice short-ranged interactions are driving the dynamics of a system. GRO-MACS uses this assumption to split a system into different domains by taking the non-bonded interactions into account, that are spatially related. The domain decomposition approach makes it possible to parallelize the computations and even further, offload specific tasks to a Graphics Processing Unit (GPU).

In order to speed up the simulation of pair-wise electrostatic and Van der Waals interactions, GROMACS uses the particle mesh Ewald approach, to handle the long-range electrostatics interactions transformed into Fourier space. With this approach the time complexity can be reduced from $\mathcal{O}(N^2)$, where $N$ is the number of particles, to $\mathcal{O}(N \log(N))$ [80].

**Machine learning.** Machine learning is an upcoming or already hot topic in research, but not tightly bound to bioinformatics. Of course, there are biological questions, that could make use of machine learning algorithms and methods. Possible data would be medical images (X-ray photographs) or other clinical related data in the topic of personal medicine. In general, machine learning can be described as a collection of algorithms improving a statistical model based on appropriate training sets or unknown data.

**TensorFlow**. TensorFlow [81] is one of the most widely used open source frameworks to implement machine learning algorithms. In this work TensorFlow is used to build a model of a convolutional neural network (CNN) based on the CIFAR-10 training model application [82] and the related CIFAR-10 dataset for image recognition. The aim of the generated model is the correct classification of images into the categories of airplane, automobile, bird, cat, deer, dog, frog, horse, ship and truck. The images are of type RGB and have a size of 32x32 pixels. The created model is build on a multilayer architecture including alternating convolutions and nonlinearities. The final classification is performed by a softmax classifier.

The time complexity of the training and testing procedures of a CNN is affected by six different parameters, the index of a convolutional layer, the number of such layers (depth), the number of filters (width) and their spatial size (length), the number of input channels, and the spatial size of the output feature map, where the number of the convolutional layers and the number of applied filters contribute mostly to the total runtime [83].

## Datasets

The used benchmark datasets are hand picked and come along with *BOOTABLE*. The different datasets were chosen regarding their runtime, to provide the possibility of rather short benchmarking runs up to longer runs. The used datasets are real scientific datasets and not created artificially. A list of the provided datasets is shown in Table 3, followed-up by Table 4, listing the datasets grouped by their runtime categories (short, middle, long). In general the shorter runs are performed with datasets that contain a smaller number of sequences than the datasets for middle runs and the middle datasets contain less sequences than the datasets used for the long runs. Except for the long runs using the full human genome dataset (GRCh38), which has a rather small amount of sequences but these are very long, also reflected by its large disk size. Since the runtime of the used tools strongly depends on the input size (number of sequences or number of bases) it is possible to regulate the runtime via the dataset sizes. For the tools GROMACS and CIFAR-10 always the same datasets are used but the run time is adjusted by different step sizes. However, it can not be excluded in general, that smaller datasets regarding their input size, may cause longer runtimes than larger ones due to their inherent complexity. In the following the various datasets are presented in detail, grouped by their application area.

**Table 4. Benchmark datasets assigned to the according tool and dataset parameters (Short, Middle, Long) of *BOOTABLE*.**

| Tool | Short | Middle | Long |
|---|---|---|---|
| BBMap | DRR001012.fa | DRR001025.fa | GRCh38 |
| Bowtie2_build | DRR001012.fa | DRR001025.fa | GRCh38 |
| Bowtie2_align | ERR016155.filt.fastq | ERR016155.filt.fastq | ERR251006.filt.fastq |
| BWA | DRR001012.fa | DRR001025.fa | GRCh38 |
| Velvet | ERR016155.filt.fastq | ERR016155.filt.fastq | ERR251006.filt.fastq |
| IDBA-UD | SRR741411.filt.fa | ERR015528.filt.fa | ERR251006.filt.fastq |
| SPAdes | ERR016155.filt.fastq | ERR016155.filt.fastq | ERR251006.filt.fastq |
| Clustal Omega | wgs.ANCA.1_200.fsa | wgs.ANCA.1_400.fsa | wgs.ANCA.1_500.fsa |
| MAFFT | wgs.ANCA.1_200.fsa | wgs.ANCA.1_400.fsa | wgs.ANCA.1_500.fsa |
| SINA | OE-38_R1.fa | RefSeq-RDP16S_v2_May2018.fa | GTDB_bac-arc_ssu_r86.fa |
| GROMACS | 10000 | 30000 | 50000 |
| CIFAR-10 model build | 1000 | 2500 | 5000 |

**Reference**. In order to use Bowtie2, BWA or BBMap to align a sequence, the index structure of the reference genome has to be build first. This step is computationally expensive and depends on the input size of the reference sequence. For Bowtie2 and BBMap the largest of the three datasets (category long) is the full human genome (*GRCh38_full_analysis_set_plus_decoy_hla.fa*, ftp://ftp.1000genomes.ebi.ac.uk/vol1/ftp/technical/reference/), taken from the 1000 Genomes Project [84]. For the category middle and short the datasets *DRR001012.fastq* (ftp://ftp.sra.ebi.ac.uk/vol1/fastq/DRR001/DRR001012/), *DRR001025.fastq*(ftp://ftp.sra.ebi.ac.uk/vol1/fastq/DRR001/DRR001025/) are used. These are taken from the sequence read archive [85], hosted by the European Bioinformatics Institute (EBI). For BWA the same datasets have been used but the index structure has been provided directly as the index creation of BWA is not multithreaded and would only contribute constantly to the scaling results. For Bowtie2_align the same dataset is used for the categories short and middle but against different references, generating different kind of runtimes.

**Assembly**. The integrated assembly applications Velvet, IDBA-UD and SPAdes use four different datasets, all extracted from the 1000 Genomes Project. The largest one is the dataset *ERR2510006.filt.fastq* (ftp://ftp.1000genomes.ebi.ac.uk/vol1/ftp/phase3/data/HG00110/) used by all three tools. Due to incompatibilities of the dataset *ERR016155.filt.fastq* (ftp://ftp.1000genomes.ebi.ac.uk/vol1/ftp/phase3/data/HG00125/) with IDBA-UD the datasets *ERR015528.filt.fastq* (ftp://ftp.1000genomes.ebi.ac.uk/vol1/ftp/phase3/data/HG00106/), *SRR741411.filt.fastq* (ftp://ftp.1000genomes.ebi.ac.uk/vol1/ftp/phase3/data/HG00099/) are added especially for it. *ERR016155.filt.fastq* is used for both, the short and middle categorized runs as the experienced runtimes fit to both categories.

**MSA**. The MSA dataset for Clustal Omega and MAFFT is taken from the NCBI hosted Genbank repository [86]. Even after longer research for the different runtime categories no matching datasets could be found that fit perfectly. Therefore, the original dataset *wgs.ANCA.1.fsa_aa* (ftp://ftp.ncbi.nlm.nih.gov/genbank/wgs/A/wgs.ANCA.1.fsa_aa.gz) was taken and the first 32, 71 and 96 sequences are used as input for the categories short, middle and long. Care was also taken to keep the file structure intact. For SINA different datasets are used, as SINA is specialized on rRNA sequences. As reference index the recommended dataset of the small subunit (SSU) Ref NR 99 in version 138.1, provided by the SILVA rRNA database project, is used. As this is the only reference dataset that is used, the index structure was computed in beforehand as the contribution to the total scaling results would have been constant.

For the alignment benchmarks of SINA, two datasets (categories middle (https://zenodo.org/record/2541239/files/RefSeq-RDP16S_v2_May2018.fa.gz) and long (https://zenodo.org/record/2541239/files/GTDB_bac-arc_ssu_r86.fa.gz)) have been taken from the DADA2 project (10.5281/zenodo.2541239). The dataset belonging to the category short (https://zenodo.org/record/803376/files/OE-38_R1.fastq.gz), has been taken from the Orchard project [87].

**GROMACS**. As dataset for GROMACS, also proposed by the developers, a part of the alcohol dehydrogenase enzyme embedded into water within a cubic box is used (https://ftp.gromacs.org/pub/benchmarks/). In order to produce longer runtimes, the step numbers are increased starting from 10,000 (short), 30,000 (middle), up to 50,000 for a long benchmark run.

**Machine learning**. For the CIFAR-10 model build application the CIFAR-10 dataset is used. The dataset is provided by the Canadian Institute For Advanced Research and is one of the standard image sets related to machine learning and image recognition (https://www.cs.toronto.edu/~kriz/cifar.html). It consists of 60,000 colored images with a size of 32x32 pixels, grouped into 10 classes, with 6,000 images per class. The CIFAR-10 dataset is used for all runtime categories. The runtime is controlled via the used step sizes of 1,000, 2,500 and 5,000.

## Results

The performance measurements are divided into two parts. The first part covers the average wall-times of the specified tools, whereas the second part presents the scaling behavior results.

### Performance measurements of virtual environments

The performance of the different environments was evaluated, measuring the wall-time of every tool from the start of the execution to its end. The wall-time also has been taken for the evaluation of the scaling performance. The choice of a measure should reflect the results as good as possible and as intuitive as possible. From a user's point of view, the most likely question is, how long will it take to get the results? The wall-time would be one of the most interesting and simple units. As the mass of numbers and results would be overflowing, a selection of the relevant results and averaged values is presented. In the following the measurements of the bowtie2-build application are neglected as the creation of the index structure involves random starting points, which can lead to larger differences regarding the wall-times. To make the results more comparable across all datasets and environments, the measurements are ignored if not noted otherwise. Further, we experienced that not all CPU core numbers can be used with GROMACS, reasons for that are discussed in the Discussion and analysis section, also the failed runs are excluded from the performance measurements of the specific CPU configurations, as their calculated percentage values would lead to a bias in the overhead results.

First of interest was to determine the overhead of the different environments (bare-metal, VM, Docker), according to the different sized datasets (short, middle, long). To gain insight into this we transformed the raw wall-time values into percentage values, where a part of these values is shown in Table 5. The final overhead values for every computing environment and dataset are presented in Table 6 and were calculated as following. First, the relative overhead of every tool for a given dataset and core configuration, between the bare-metal environment and one of the virtual computing environments (VM or Docker) is calculated from the raw wall-time values. Afterwards the average percentage for every core configuration over all tools is calculated to get the per core configuration overhead stats. Finally, the average per core percentages are summed up and divided by the number of core configurations, excluding the not ran GROMACS benchmarks for special CPU configurations. This was done for all environments, combined with the different CPU core configurations.

**Table 5. Example of the summed up and averaged percentage values for the short dataset of the used envrionments (bare-metal (BM), virtual machine (VM), Docker, Docker on bare-metal).**

| Environment | Overhead [%] | | | |
|---|---|---|---|---|
| | **1 core** | **2 cores** | **...** | **36 cores** |
| BM-VM | 7.71 | 8.20 | ... | 9.67 |
| BM-Docker | 12.15 | 20.43 | ... | 23.75 |
| BM—Docker on bare-metal | 9.02 | 10.54 | ... | 4.65 |
| VM-Docker | 4.25 | 11.89 | ... | 13.08 |

**Table 6. Average overheads over all used CPU core numbers of VM and Docker runs and one run of Docker directly on bare-metal.** The overhead numbers are related to the bare-metal benchmarks, expressed in percentages, including the used dataset.

| Environment | Dataset | Overhead [%] |
|---|---|---|
| VM | short | 7.73 |
| Docker | short | 22.54 |
| Docker on bare-metal | short | 6.63 |
| VM | middle | 10.51 |
| Docker | middle | 24.45 |
| VM | long | 7.12 |
| Docker | long | 14.94 |

The final results show that virtualization on VM level leads in average to an overhead of around seven to ten percent, related to the bare-metal values as reference. Regarding Docker, the overhead is even higher than for the VM with 15 to 24 percent, compared to the bare-metal values. We can also see that the increase of the used datasets in size and complexity, leads to nearly the same overhead results with deviations in the range of around three percents. If the raw time values are illustrated as a stacked bar plot (shown in Fig 2), more details regarding the used CPU numbers are revealed. One specific detail are the wall-time values of the bare-metal single core performance. The values are lower than the ones of the virtual environments. The single core performance of a VM, for example using the middle dataset, is around 13 percent lower, meaning the benchmarks inside a VM needs 13 percent more time for the same amount of work compared to a bare-metal machine. But with the increasing number of cores, the overhead values of the VM environment decreases, down to the average values. For the Docker environment inside a VM the overhead is nearly constant in the range of around 20 to 25 percent, a behavior with decreasing values have not been observed, over all three dataset categories.

Taking a closer look at the differences between the results of the VM environment and Docker, some cases were found, where the measured wall-time of a virtual environment was lower than for the bare-metal environment. It also occurred that the Docker environment required less time than runs on a virtual machine. These cases have been observed for all datasets but it occurs sporadically for different tools and different core configurations in the range of two percent with some rare outliers of ten to twelve percent, mostly related to Clustal Omega. Furthermore, a benchmark of the short dataset, using Docker on bare-metal, has been conducted for the sake of completeness. The results show, that the difference of the wall-times between the bare-metal environment and Docker on a bare-metal environment can be compared to the one of the VM environment. The exemplarily conducted run of the *BOOTABLE*

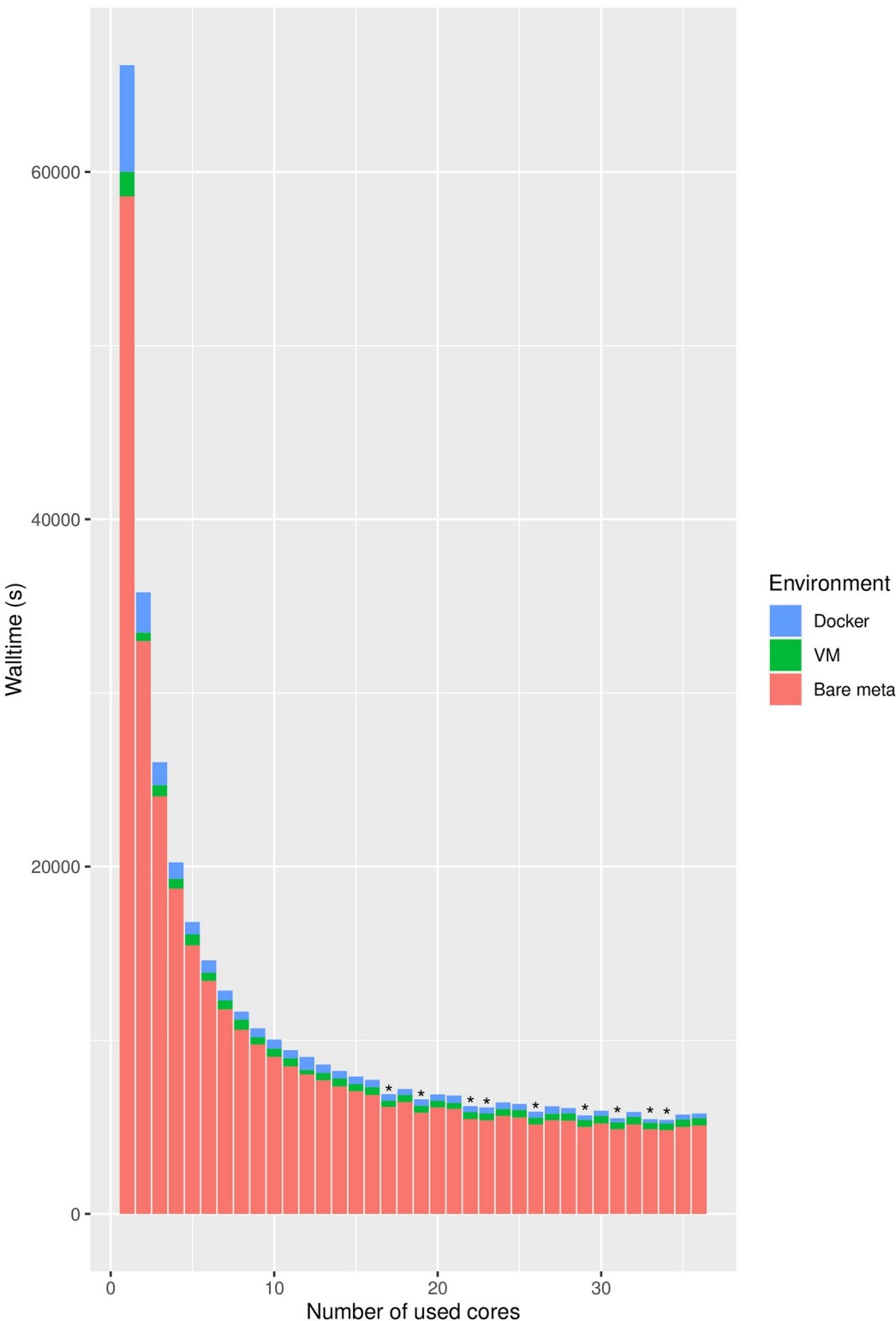

**Fig 2. Wall-time comparison of virtualization environments.** CPU core number-wise comparison of summed up wall-times for the different computing environments based on the long dataset, presented as a stacked bar chart. The stacked parts needs to be seen as addition to the already existing part under it. Red bars represent the results of the bare-metal environment, green the additional time in a VM and blue, using a Docker container. The bars marked with a star indicate CPU core numbers, where GROMACS did not allow to start the simulation.

Docker container showed the same behavior as the VM environment with a similar overhead value of around seven percent.

## Scaling behavior

In this section, the focus is shifted towards the scaling behavior of the different tools. In order to visualize the discovered results in tabular format, the scaling behavior of the tools are binned into one of three categories, yes it scales, no it does not scale and partially, if the behavior can not be sorted into one of the categories yes or no. The results are illustrated in Table 7.

To get a direct impression if a tool scales, ideally linear using additional CPU cores or not, we decided to create scaling plots, using logarithmic x and y axes. In the following the scaling plots of the long dataset in a bare-metal environment are presented. The scaling plots of the different environments do not differ significantly from each other. The same applies to the different datasets, they only differ in the scale of the measured wall-times. A first look on the scaling plots of the sequence alignment tools, presented with Fig 3, shows a quite diffuse behavior for bowtie2_build (top left). Especially for higher numbers of CPU cores, the values are alternating. An explanation of this behavior is given in the Discussion and analysis section. For bowtie2_align (top right) a nearly perfect linear graph is shown, which points towards a linear behavior for the alignment process. For the short and middle dataset the scaling is not that perfectly linear but still satisfactory. For BWA using the MEM algorithm also a mostly linear behavior is shown (bottom right), except for the last core configuration using 36 cores, showing an unusual wall-time increase. The linear behavior from the alignment tools before, also applies to the third alignment tool, BBMap (bottom left). The scaling is close to linear, but with less improvements towards the maximal number of available cores for the short dataset. In general all selected tools of the sequence alignment category showed a sufficient scaling performance.

The results of the *de novo* assembly tools are shown in Fig 4. The *de novo* assembler Velvet is split into two execution steps and considered as two separate tools (Velveth, Velvetg). On the first look Velveth shows a quite diffuse behavior, going up and down, but the range of the lowest and the highest measured wall-time value is in a range of 1 second and therefore nearly

**Table 7. Classification of the used applications by the scaling results into one of the three different categories Yes, No and Partially, related to the long dataset.** The long dataset also represents the datasets short and middle. Category Yes means, that the application scales satisfactory with the number of used cores. Category No means it does not scale. Partially means, that for some numbers of CPU cores the application scales satisfactory for other not.

|  | Yes | No | Partially |
|---|:---:|:---:|:---:|
| BBMap | ✓ | - | - |
| Bowtie2_build | ✓ | - | - |
| Bowtie2_align | ✓ | - | - |
| BWA | ✓ | - | - |
| Velveth | - | ✗ | - |
| Velvetg | - | ✗ | - |
| IDBA | - | - | ✓ |
| SPAdes | ✓ | - | - |
| Clustal Omega | - | ✗ | - |
| MAFFT |  |  | ✓ |
| SINA | ✓ | - | - |
| TensorFlow | ✓ | - | - |
| GROMACS | ✓ | - | - |

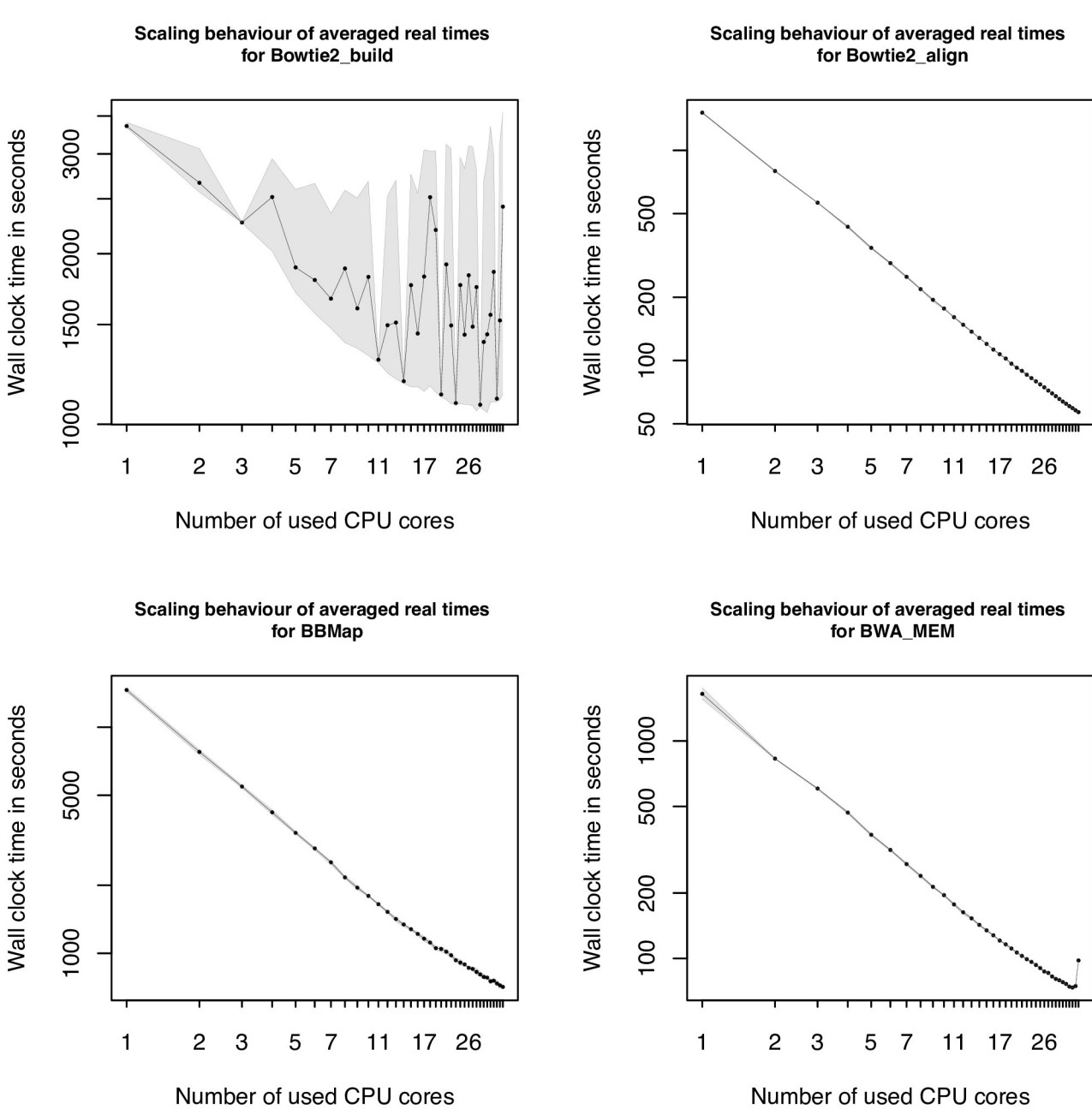

**Fig 3. Scaling plots 1—4.** Graph, representing the scaling behavior of Bowtie2_build (top left), Bowtie2_align (top right), BBMap (bottom left) and BWA (bottom right), using average values from five replicas from the long dataset. Both axes are scaled logarithmically, to make a linear behavior optically visible. The values on the x-axes represent the number of used cores, whereas the y-axes show the measured wall-time in seconds. Please note, that the y-axes of the different scaling plots cover different ranges of measured wall-times. In addition to the curve, the borders of the gray area show the minimal and maximal wall-time values measured.

constant, according to the used CPU cores. Velvetg shows a similar behavior as Velveth but in a range of around ten percent between the minimum and maximum values. The bottom left plot of Fig 4, shows the scaling results of the assembly tool IDBA. In the beginning, the wall-times scale almost linearly regarding the used CPU cores, until 16 cores. Afterwards the values stagnate and the graph flattens out. With increasing core numbers, towards the maximal number of 36 cores, the wall-time values even increase. The last tool of the assembly category is

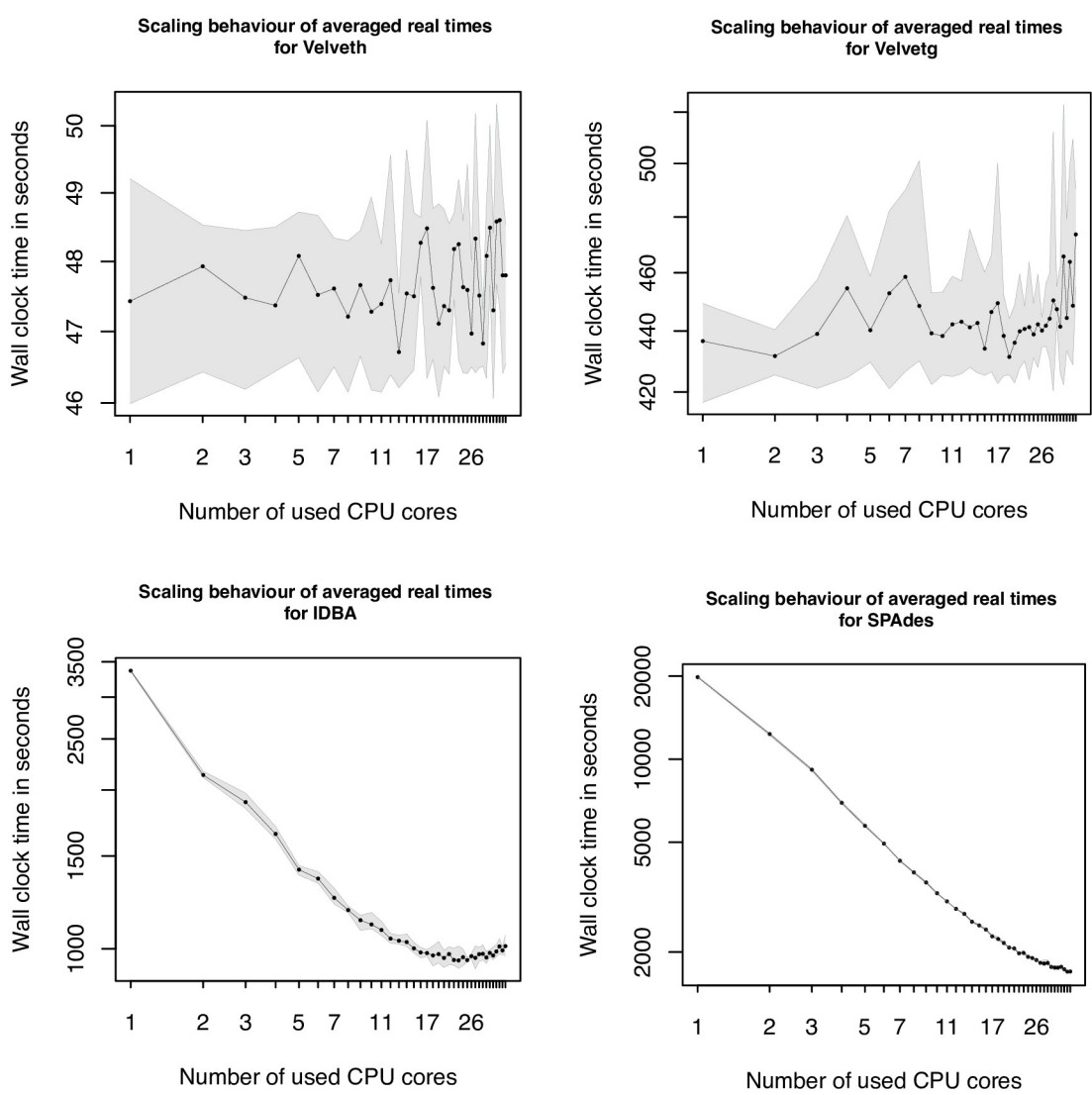

**Fig 4. Scaling plots 5—8.** Graphs, showing the scaling behavior of Velveth (top left), Velvetg (top right), IDBA-UD (bottom left) and SPAdes (bottom right) of the long dataset. The x-axes represent the number of used cores in the range of 1 to 36. The y-axes show averaged wall-time values in seconds, in different ranges, depending on the tool. Both axes are scaled logarithmic to make a linear behavior visible. In addition to the scaling graph, the minimal and maximal values of the five conducted runs are illustrated by the borders of the area in gray.

SPAdes. The scaling graph is shown in the bottom right. SPAdes has the highest runtime compared to the other assembly tools (Velvet, IDBA-UD). The graph shows an almost linear scaling behavior, with a light bump for two cores and a light flattening, starting from 17 cores on.

The scaling behaviors of the MSA tools are illustrated in Fig 5. The graph on the left shows the behavior of Clustal Omega. It reveals a contrary view on the scaling. The lowest wall-time values are achieved, using only one core. The more CPU cores are used, the higher are the wall-time values. With the last configuration, the usage of 36 cores, the total wall-time value has almost doubled. MAFFT shows a different scaling behavior than Clustal Omega. Up to 8 cores it scales almost linear with smaller up and downs. Beyond the number of 8 cores, the wall time values stagnate and alternate slightly. The tool SINA, specialized on rRNA, shows a

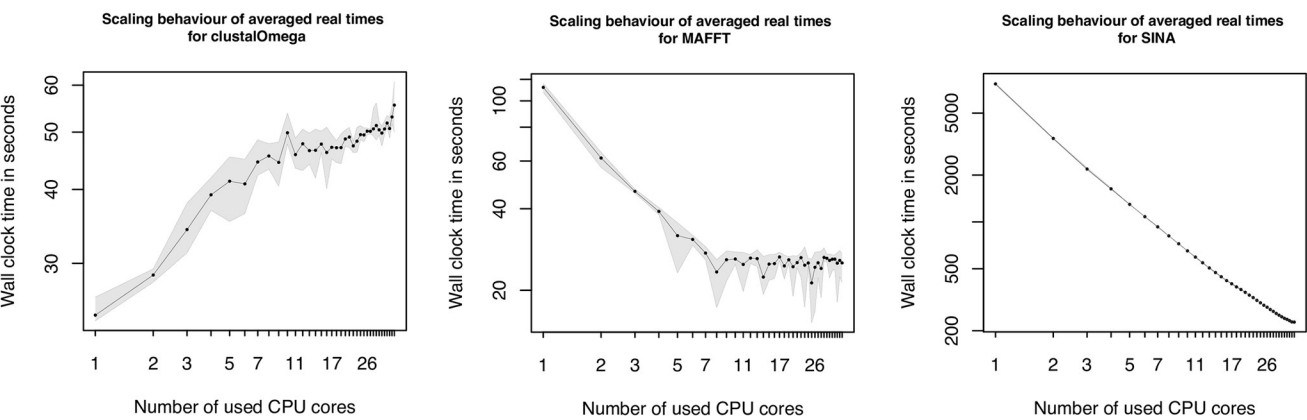

**Fig 5. Scaling plots 9—11.** The graphs illustrate the scaling behavior of Clustal Omega (left), MAFFT (middle) and SINA (right) using the long dataset. The x-axes values are in the range of 1 to 36 and represent the number of used cores. The y-axes show averaged wall-time values in seconds. The ranges of the y-axes differs, depending on the tool. Both axes are logarithmic, to make a linear behavior visible. To indicate the minimum and maximum wall-time values of the five executed runs, a gray area has been inserted, whose boundaries represent the lowest and highest measured values.

very linear scaling behavior (graph on the right in Fig 5), with slightly less decreasing wall-time values for higher numbers of CPU cores.

The fourth series of scaling graphs (see Fig 6) shows the scaling behavior of the CIFAR-10 TensorFlow application and GROMACS. The TensorFlow application scales practically linear with a light bend after ten cores and a very light flattening effect at the end. For the molecular dynamics simulation tool GROMACS a rather odd behavior can be observed for the CPU core numbers 17, 19, 22, 23, 26, 29, 31, 33 and 34. For other numbers of CPU cores the graph shows a regular behavior, regarding the measured values from the lower CPU core numbers. The reason for this behavior is clarified in the Discussion and analysis section.

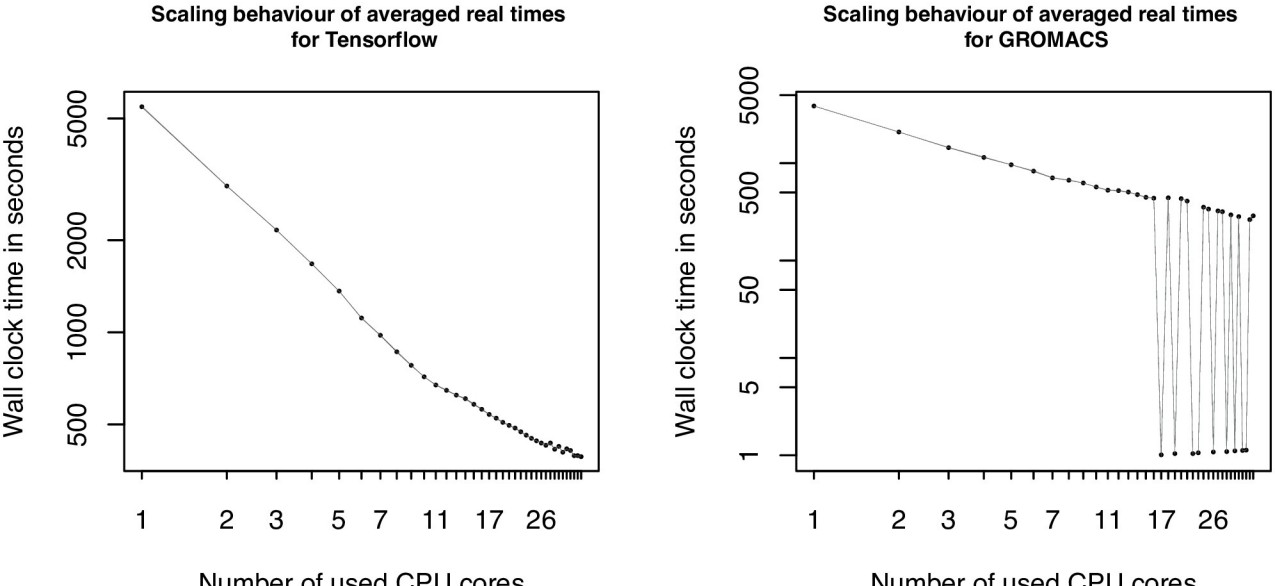

**Fig 6. Scaling plots 12—13.** Scaling graphs showing the TensorFlow application CIFAR-10 (left) and GROMACS (right) using the long dataset. The number of used cores are represented on the x-axes, whereas the y-axes show the averaged measured wall-times, in different ranges, depending on the tool. The x and y axes are scaled logarithmically to make a linear scaling behavior visible. The borders of the gray area illustrate the minimal and maximal measured values for each used core, but are barely visible as only tiny deviations from the different replicas have been measured.

Further explanations and discussions of the observed scaling behaviors can be found in the following Discussion and Analysis section.

## Discussion and analysis

The Analysis and Discussion Section is divided into two parts. Each part discusses the found results and gives explanations for the observed behaviors.

### Virtual environment comparison

The obtained results show, that the different virtualization environments affect the measured wall-times almost as expected. The bare-metal setup performs best, afterwards come the VM benchmarks and finally the Docker environment. This was expected as the hypervisor layer (KVM), required for the virtualization, adds an additional layer of abstraction to the environment and therefore a loss of performance is observed. The overhead can be specified with around seven to ten percent over all datasets for the VM environment, compared to the performance of the bare-metal setup. The performance of the benchmarks, using Docker, are worse compared to the VM setup and thus also worse than the bare-metal performance. Over all datasets the overhead is more than 15 percent compared to the bare-metal values. The overhead between the VM and Docker environments is rather high with around 8 to 15 percent on top of the VM performance. The additional loss of performance has been expected, as Docker containers were run on top of the hypervisor layer inside a VM. The loss of performance can be explained by two reasons. Besides the additional virtualization layer of the Docker engine, the used file system of the executed container is responsible for a light performance loss. We used the recommended overlay2 driver, but still there exists some overhead. This also seems to be the reason for the larger overhead, compared to the VM environment. A good example to show the additional overhead, created by the additional layers, is the CIFAR-10 application using the TensorFlow framework. TensorFlow is mostly written in C and therefore it makes lots of system calls. Due to the high number of system calls, the computations executed through the TensorFlow framework also spend a larger amount of time for these system calls. Executed in a bear metal environment the system calls are faster, than on a VM or inside a container, as there is no need to translate the system calls from the guest operating system to the host and back. Based on the measured wall-time values of the CIFAR-10 tool, this can result in an overhead of 25 to 30 percent. Further, evidences supporting this assumption are provided by GROMACS, that shows a similar system call behavior like the CIFAR-10 tool and similar overhead results, regarding the different execution environments.

The additional benchmark run of the dockerized version of *BOOTABLE* directly on bare-metal showed, that the wall-times are close to the measured values of the VM environment. The performance of Docker on a bare-metal instance was not evaluated beyond the short dataset, as there are hardly any major infrastructure operators that allow such a setup, due to security reasons. But for smaller infrastructures it might be of interest, so this scenario has been taken into account.

Furthermore, we expected to see different behaviors of the tools regarding the different datasets. Of course the larger and more complex datasets showed an increased wall-time but apart from that we only recognized some odd behaviors regarding the scaling behavior of Clustal Omega, which is discussed in the upcoming Scaling behavior section.

### Scaling behavior

After investigating the various scaling plots we cannot see any effects on the scaling behavior, triggered through the different execution environments. The same applies to the different

datasets, which also have no or only a very small effect on the scaling performance. The only difference is observed for IDBA-UD which produces a flat graph for higher CPU core numbers on the long dataset instead of a rising graph for the short and middle dataset. This behavior can be observed for all execution environments.

The scaling performance of the Bowtie2 build tool does not look like it would scale, but taking a closer look on the raw wall-time data reveals some fluctuations. The values used for the generation of the scaling plots are the mean of five benchmark runs for every CPU core configuration. Some of the five replicas showed significant shorter wall-times, but other higher ones. The reason for this is, that Bowtie2 chooses a random starting point based on the random number generator seed. Depending on the generated random number, the runtime can increase by factors of 2 or 2.5. This of course affects the calculated average values. If only the lowest values would be taken, ignoring the outliers, the graph would show an almost linear scaling behavior. The Bowtie2 aligner instead shows a nearly perfect linear scaling behavior, with each CPU core doubling almost halving the measured wall-times. This is also true for the other tools of the sequence alignment tools category, BBMap and BWA. For BWA an increase of the wall-time, using 36 cores have been seen, which might be related to operating system processes that also require some CPU time. In general, these results have been expected as the alignment of two sequences is a well-known and studied problem in bioinformatics, that can efficiently be solved using a dynamic programming approach. This approach is applicable because the problem can be divided into smaller subproblems (partial alignments), which can be processed independently of each other. This makes it possible to distribute the load efficiently on the available CPU cores.

A more complex problem in bioinformatics is the alignment of multiple sequences, covered by the applications Clustal Omega, MAFFT and SINA. Clustal Omega provides a multithreading option, but during our study we observed the best performance, using a single core. Furthermore, the provided graph in Fig 5 shows almost continuously increasing values for higher numbers of used CPU cores. The results are also not comparable to the one of Velvet, where the usage of multiple compute cores do not lead to a significant speed up. In the case of Clustal Omega it leads to a doubling of the runtime, comparing the values of the best performance for one core to the worst using 35 or 36 cores. Also there is no sweet spot, where the wall-time would first decrease and after a specific number of cores would increase. If switched to the verbose mode of Clustal Omega the output reveals, that the longest taking part is the progressive alignment step which seems to be affected negatively by the usage of multiple cores, if datasets with a small number of sequences are used. From the documentation of Clustal Omega we can see that the procedures, to calculate the pairwise distances and parts of the HMM building procedures are parallelized. We can approve, that these parts benefit from the additional, cores but the subsequent progressive alignment step negates the achieved benefit. To verify our assumption that the small sizes of the datasets are responsible for the resulting overhead, we performed an additional test on the full ANCA.1 dataset. The first information we could gather from the provided output is that only from a number of more than 100 sequences, the mBed algorithm is used to calculate the pairwise distances. The second finding was, that a test run on the full dataset (1258 sequences), using a single CPU core, took two times longer, than the same dataset using the maximum of 36 cores. But using only 19 cores leads to an even better performance, which seems to be the sweet spot in this case. Using more CPU cores did not lead to a further reduction of the measured wall-time, on the contrary, the numbers slightly increase. The full scaling plot is shown in Fig 7. The second representative of the MSA category is MAFFT. The observed behavior could be caused on the one hand by the used algorithm itself or on the other hand by the implementation. Both could lead to a limitation of the scalability to 8 CPU cores. In contrast to Clustal Omega and MAFFT, the third MSA tool, SINA,

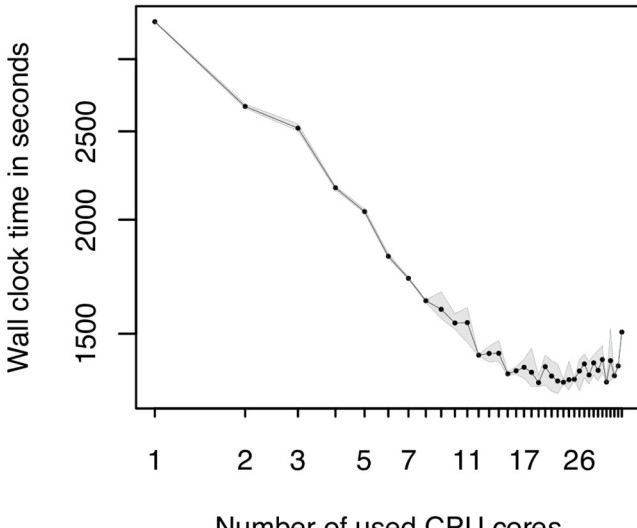

**Fig 7. Scaling plot for larger Clustal Omega dataset.** Additional scaling plot showing the scaling behavior of Clustal Omega, using the full ANCA.1 dataset. The x-axis represents the number of used cores, where the y-axis illustrates the measured wall-time in seconds. Due to the logarithmic x and y axes, a linear scaling performance can be made visible. The borders of the gray area show the measured minimal and maximal wall-times from the multiple replicas.

scaled very well. The reason for this might be the different setup and the usage of a reference index, that would put SINA in between the sequence alignment category and the multiple sequence alignment category.

For two of the three *de novo* assembly tools a rather poor to medium scaling performance has been observed. Although Velvet offers multithreading, most of the computations are not parallelized and therefore most of the workload is done using a single core. With that knowledge, it is not surprising that the scaling plots show some diffuse performance value fluctuations, but in a rather small range. Therefore, it does not make a large difference, if more than a single core is used. The second *de novo* assembly tool is IDBA-UD. It also offers multithreading and the performance measurements show, that it makes use of the parallelization in most parts of the algorithm. The determined change in the scaling behavior, if more than 16 CPU cores are used, let assume that the implemented parallelization procedures are somehow limited to this number or it is not possible to improve the parallelization because of the algorithm. To the best of our knowledge, there seems to be no reason why the algorithm should not make use of more than 16 compute cores, so the limiting factor might be the implemented parallelization procedures.

The third tool belonging to the category of *de novo* assembly tools, SPAdes, scaled best among the *de novo* assembly tools but also showed a good scaling performance, compared to all other benchmarked applications. In the beginning the runtime is almost halved by doubling the cores, but this behavior stops with the step from 8 to 16 cores. But still the runtime can be decreased with each core by two to three percent as observed for the long dataset on bare-metal machine. There is still a gain in performance, also for higher CPU core numbers that the other assembly tools in this study do not show. In general, SPAdes shows high wall-time values. This can be explained by the design of the algorithm, which iterates over different sizes of k-mers. As SPAdes uses k-mers with different lengths to optimize the assembly, each step

needs nearly the same amount of time. Reducing the number or range of $k$ would lead to lower wall-times. As we did not put any effort on the evaluation on the quality of the results, we left the decision for the chosen $k$'s to SPAdes.

TensorFlow or more precise the CIFAR-10 application showed a scaling behavior mostly as expected. The training workload can be easily divided as the intermediate computations are not depending on each other. Further, the TensorFlow framework is tuned to a high performance. Nonetheless, in the beginning the wall-time values are almost halving. With the usage of more than ten cores this behavior stops. The runtime is still decreasing but not as strong as before. This behavior is shown over all environments and all numbers of applied steps. So far we were not able to find a plausible explanation for that behavior but as this has been observed independently of the environment and the size of the applied steps, we assume that something on the TensorFlow or CIFAR-10 code level might be a reason for this.

The most noticeable behavior shows GROMACS. Up to a usage of 16 CPU cores everything is fine and the runtimes are almost halved by doubling the number of cores, leading to a good scaling performance. Unfortunately, there are inconsistencies, interrupting the linear trend, caused by measured runtimes of around one second, meaning that GROMACS did not start the simulation. The log files reveal that the chosen number of CPU cores can not be applied to the present molecule due to the resulting domain decomposition. The reason for that behavior is, that the GROMACS developers have put lots of effort into performance tuning. A core piece is the built-in domain decomposition algorithm, to use dynamic load balancing. The following explanation was deliberately kept very short and simple in order to not exceed the scope of this work. A more detailed description can be found in the original paper [88] and [89] for the updated version. The molecule, or more precise the simulation box, is divided into different cells. These cells need to communicate with each other, based on the cutoff radii for bonded and non-bonded interactions. This means if cell 0 is the starting point, it will be checked, which other cells are interacting with cell 0 and need to communicate with it. These interactions defines how the simulation load is distributed on the processors. The behavior observed in our study can be explained by the fact, that all of the failed CPU core configurations are larger prime numbers by itself like 29 or consists of larger ones like 34 (2 times 17). Depending on the molecular system, such prime numbers chosen for the number of threads, can end in very sharp and thin cells, resulting in problems, simulating interactions correctly. Therefore, GROMACS performs a check, if the chosen thread number is appropriate or not. If not the simulation will be stopped at the beginning, resulting in the measured one second wall-times. For other molecular systems than the one used in this study, this can change and the non working CPU core configurations could work. However, the newer versions of GROMACS no longer use the concept of charge groups. They have been replaced by pair lists using a Verlet buffer containing the interactions of particles, or more precise particle clusters, that are constructed by spatial gridding in two dimensions and spatial binning in the third dimension. The outcome remains the same, using prime numbers as thread numbers can lead to very thin and sharp cells or very large cells with too many interactions, that could not be simulated efficiently.

## Generalizability

Throughout this work several aspects of the scaling behavior of the different applications, datasets and execution environments have been discussed, including the possible overhead of the different environments. First, the presented results are valid for the selected tools, datasets and environments. However, for specific findings a general statement can be made. Looking at the different datasets and the resulting runtimes, no changes regarding the scaling behavior were

noticed. Therefore, the size of a dataset seems to have no effect on the multithreading abilities of a tool, assuming a tool behaves the same, independently of the input size, as seen for Clustal Omega, where parts of the algorithm are only activated if a certain input size threshold is passed. But in general it seems to be true that datasets have no influence on the scaling behavior. With this knowledge, it would be possible to run performance evaluations on smaller datasets first and save time on larger ones. The same statement can be made, related to the different environments. There seems to be no effect on the scaling behavior, but of course an overhead produced by additional layers, will always exist. Another result that can be transferred to other tools and applications are results regarding the used programming language and the corresponding implementation. As already mentioned in the environment comparison section, for the CIFAR-10 TensorFlow application and GROMACS, the performance difference between a bare-metal environment and a virtual environment is rather high, compared to other tools, as the implementation of both tools is based on programming languages (C or C++), that are able to map their instructions efficiently to machine instructions. This benefit will get lost if one or more translation layers are in between. Further, it is possible to generalize this result. The closer a tools is implemented to machine instructions, the higher is the performance loss if virtual environments are used, especially with regards to the single core performance.

A generalized statement, based on algorithms or data structures is more difficult. Algorithms are used to solve a given problem or to find an approximate solution. The problems to be solved are different for the selected tool categories. For the category of sequence alignment all tools showed a pleasant scaling behavior. The same is true for the CIFAR-10 application and GROMACS. All the underlying problems of these ones can be parallelized very well, especially for the category of sequence alignment. For the multiple sequence alignment category it looks different, also for the assembly tools. In both categories, tools are struggling to show a scaling performance close to linear. This suggests, that due to the nature of the problem, paralellization is difficult to achieve. Even if similar data structures are used, like for the selected assembly tools (Velvet, IDBA, SPAdes). For Velvet only small parts of the implemented algorithm are parallelized. For IDBA the benefit of multiple cores stops with 16 cores, whereas SPAdes shows a satisfying scaling behavior. Based on the data structure only, no general conclusions can be made, it still depends on the algorithm and the implementation. A look on tools using similar algorithms like Bowtie2 and BWA or MAFFT and Clustal Omega reveals, that some general assumptions can be derived. Both pairs of tools behave similar regarding their scaling properties, especially Bowtie2 and BWA are very close in terms of their algorithms and data structures. Therefore, it is possible that tools behave the same but in the end it still depends on the implementation.

## Conclusions and future work

The tested tools behaved almost as expected in the different runtime environments and also regarding the different datasets. Based on the collected results it can be said that virtual environments cause an overhead that can not be neglected. It needs to be decided from tool to tool if the valuable flexibility makes it worth to loose some performance. Tools designed to make use of bare-metal hardware features might experience a stronger performance loss, than tools implemented in a more abstract design. So it needs to be decided from application to application, whether a generated overhead is acceptable but linked to a gain of flexibility or the other way round. Paralellization is a good way to speed up processes but unfortunately not every of the tested tools, offering parallelization through multithreading, scales with larger numbers of CPU cores. It depends strongly on the underlying problem, if it can be divided efficiently into

smaller subproblems or not. A further strong limitation can result from the implementation of multithreading libraries as our study showed. Therefore, algorithm and implementation must fit well together.

One outcome of this study is, that the required time to generate the index structure of a reference genome with Bowtie2 can have large differences. Since a reference index structure is usually built only once, most users will probably not even notice. Instead the pairwise alignment procedure, also performed in this study by Bowtie2, shows a nearly perfect scaling behavior and therefore takes huge advantage of multiple CPU cores. This is also true for the other sequence alignment tools, BBMap and BWA. Other tools, like Velvet seem to gain no advantage from multiple available cores or are limited to a supported number like IDBA-UD. As other tools, like SPAdes, using a similar data structure scale well, there seems to be a limit due to the implementation. As mentioned first, the used tools are mostly not affected by the chosen datasets, except Clustal Omega. For Clustal Omega the number of sequences can make a difference but larger datasets should benefit from the usage of multiple cores, as the additional benchmark run showed. Tools that have been proven to be robust and very scalable are TensorFlow, the CIFAR-10 application in particular, and GROMACS. The scalability of both tools benefits mostly from a well defined decoupling of the main problem into smaller subproblems, which is a basic requirement to achieve a high scalability.

For the future we plan to extend *BOOTABLE* by more applications and cover even more research areas. Especially the topic of GPU accelerated tools in the area of bioinformatics seems to be promising to get a deeper insight, where GPUs make sense or not, also regarding virtual environments and scaling effects. Further, we want to add more datasets and provide more suggestions on the types of datasets that can have an impact on different tools, as seen for Clustal Omega. Finally, we plan to build a public database of already benchmarked tools and curated recommendations about their reasonable resource usage and constantly adding new results to make the life of users and resource providers a bit easier. It is not always the more the merrier.

## Acknowledgments

The authors thank Volker Lutz for his ideas and valuable inputs throughout the conduction of the computational tasks and analysis of the results. Further, we want to thank Johannes Werner for his suggestions regarding tools and datasets.

## Author Contributions

**Conceptualization:** Maximilian Hanussek, Felix Bartusch, Jens Krüger.

**Data curation:** Maximilian Hanussek, Felix Bartusch, Jens Krüger.

**Formal analysis:** Maximilian Hanussek, Felix Bartusch, Jens Krüger.

**Funding acquisition:** Jens Krüger.

**Investigation:** Maximilian Hanussek, Felix Bartusch, Jens Krüger.

**Methodology:** Maximilian Hanussek, Felix Bartusch, Jens Krüger.

**Project administration:** Jens Krüger.

**Resources:** Maximilian Hanussek, Felix Bartusch, Jens Krüger.

**Software:** Maximilian Hanussek, Jens Krüger.

**Supervision:** Felix Bartusch, Jens Krüger.

**Validation:** Maximilian Hanussek, Felix Bartusch, Jens Krüger.

**Visualization:** Maximilian Hanussek, Felix Bartusch, Jens Krüger.

**Writing – original draft:** Maximilian Hanussek, Jens Krüger.

**Writing – review & editing:** Maximilian Hanussek, Jens Krüger.

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
