## [Decision Letter · Decision Letter 0]

6 Jan 2021

Dear Dr. Krüger,

Thank you very much for submitting your manuscript "Performance and scaling behavior of bioinformatic applications in virtualization environments, to create awareness for the efficient use of compute resources." for consideration at PLOS Computational Biology.

As with all papers reviewed by the journal, your manuscript was reviewed by members of the editorial board and by several independent reviewers. In light of the reviews (below this email), we would like to invite the resubmission of a significantly-revised version that takes into account the reviewers' comments.

We cannot make any decision about publication until we have seen the revised manuscript and your response to the reviewers' comments. Your revised manuscript is also likely to be sent to reviewers for further evaluation.

Sincerely,

Christos A. Ouzounis

Associate Editor

PLOS Computational Biology

Edwin Wang

Benchmarking Editor

PLOS Computational Biology

Reviewer's Responses to Questions

**Comments to the Authors:**

Reviewer #1: 1- The work investigates the performance of a number of bioinformatics tools in respect to their scaling and virtual environments performance attributes using different datasets and a benchmarking tool that was developed by the authors (BOOTABLE). The comparison involved using OpenStack and Docker virtualization environments for running a number computational tools that are used in various bioinformatics applications such as: Bowtie2, Velvet, IDBA, Clustal Omega, TensorFlow, GROMACS, SPAdes using a to a bare-metal setup and among each others. The research idea is quite interesting due to the increase use of cloud computing virtualization environment to tackle issues related to the increasing volume of data and tools application scalability.

2- However, authors need to justify the selection of bioinformatics tools which they included in their comparison rather than others. The selected tools have been a representative sample for all bioinformatics tools taking into consideration their underlying computational algorithms and technologies and the bioinformatics applications covered. The authors must have also provided justification for using Docker and OpenStack virtualization tools rather than others. In addition, the authors need to provide more elaboration regarding the performance comparison criteria which they used in the study. The authors must also comment on the generalizability of their findings to other bioinformatics tools, algorithms, applications and virtualization which is quite important for the paper to be more relevant to the readership of PLOS computational biology.

3- The tools section must be rewritten to provide a clearer categorization of the bioinformatics tools. The categorization must cover the bioinformatics applications which the tools are used in and the computational technologies and algorithms which they implement. Highlighting those two aspects is extremely important for the comparison soundness, validity and generalizability.

4- I suggest categorizing and summarizing the the tools using a table or figure after adding sufficient and more adequate justification for their selection as suggested in (2). As a result, subsections in the tools section pages 7-12 must then be reorganized and rewritten. Machine learning is an AI field and should not be categorized or be referred to as a bioinformatics tool as in page 1.

5- In terms of the paper presentation. The authors needs also to make some language correction, editing and rephrasing particularly in the abstract, summary, introduction and the conclusion sections. The figures must be used more effectively in presenting the results of the comparison and their legends must be expanded to provide more details regarding their meanings. In addition, the datasets section must come before the tools section.

The research topic of the paper is interesting and the effort that was made to conduct the comparison is quite considerable. However the paper needs rewriting taking into consideration the comments 1-5 in order to be accepted.

Reviewer #2: In this work, the authors conduct an experimental study to investigate (a) the overhead added to the execution times of various widely-known bioinformatics tools when VM or container-based implementations are used instead of bare metal implementations, and (b) the scaling behaviour of bare metal, VM and containers implementations of the previous tools as more computational cores are available.

In general, the topic covered is interesting, however, the number of bioinformatics tools examined is very limited (below ten, coming from five application areas). Consequently, the experimental findings cannot be generalised and the most examined application areas are not well-represented (apart from the “sequence assembly” category, all the other ones are represented by only one tool). It would strengthen the impact of the work to cover more tools from each of the examined application areas.

Another issue is that, based on the scope of the PLOS Comp. Biology, “research articles should model aspects of biological systems, demonstrate both methodological and scientific novelty, and provide profound new biological insights”. It is not clear how this work provides new biological insights, thus I am not sure if another journal would be a better fit for it. But this is not a strong opinion from my side.

Moreover, it is very important that the authors should describe their experimental setup effectively. First of all, the last lines of the “BOOTABLE” subsection, have significant overlap with the “System environment” subsection. The authors should remove the relevant content from the former and enrich accordingly the text in the latter. Moreover, more details are needed on how the dockerized version of BOOTABLE works. Is there any reason why the authors did not try to use this dockerized version on bare metal? If yes, the authors should elaborate on this. If no, they should include experiments using the dockerized version deployed on bare metal. Finally, in the case of VMs, which was the virtual-to-physical cores ratio used? Was it a 1:1 ratio?

Furthermore, there are parts of the manuscript that are really fluffy and should be revised to be more compact. For example, since section “Tools” refers to widely known bioinformatics tools, it should be more compact. There is no need for long descriptions for each tool, a couple of sentences (along with the corresponding references) will be sufficient. Also the “Scaling behavior” subsection in the “Discussion and Analysis” section could be more brief.

Regarding the experimental section: The distinction between small, medium, and large datasets is unorthodox: For instance, in some cases, the small and medium dataset used for a particular tool are identical. This should change. In fact, most data sets are of similar order of magnitude, thus, it would make sense for the authors to leave completely behind this data set classification. Moreover, the authors mention that for all experiments they take the averages of 3 consecutive runs. First of all it would be preferable to include more measurements (e.g., at least 5, since they mention that they have fluctuations for some tools) but they should also include error bars to the figures of their experiments.

Another important issue is the following: The authors should restructure the tables in the experimental section, since their current structure is difficult to follow. For example, Tables 4, 5 and 6 provide different views on the same experiment and are confusing. First of all, Table 4 could be removed, since it contains no important information. In Table 5, the VM-Docker line could also be removed; the important is to show the overheads compared to the bare-metal approach, the differences between VM and containers can be easily discussed in the text. In this way, Table 5 will be more compliant with Table 6. Also Table 7, could be easily transformed to contain 9 columns; the current version with 3 columns is confusing without a reason.

A couple of other issues:

- In page 3 the authors refer to MapReduce and Apache Spark as “new” programming models. However, both of them have several years that have been established as industrial standards, thus it is more proper to refer to them as “modern” programming models.

- Table 1 is placed in p.12 however it is first mentioned in p.8. The authors should move it to be closer to its first mention.

In addition, the manuscript contains many typos and linguistic errors making it difficult to read, thus it should be revised by a native english speaker. An indicative list follows (however, there are many other issues in the text):

- [abstract:] “but not always is more compute power helpful” => “but more compute power is not always helpful”

- [abstract:] “in two different virtual environment” => “in two different virtual environments”

- [p.1:] “Therefore it would desirable” => “Therefore it would be desirable”

- [p.1:] “another factor are” => “another factor is”

- [p.5:] “about the terms scalability” => “about the term scalability”

- [p.5:] “like GPGPUs” => “like GPUs”

- [p.6:] “For SPAdes the same experience has been made.” => “Similar results have been observed for SPAdes.”

- [p.7:] “we used three different kind of” => “we used three different kinds of”

- [p. 17:] “the found results” => “the results found”

- [p. 17:] “time for this system calls” => “time for these system calls”

- [p. 18:] “Some of the three replica” => “Some of the three replicas”

- [p. 18] “This results have been expected” => “These results have been expected”

- [p. 20] “bare-metal hardware features might suffer stronger than tools => ??? what do you mean ??? in which way they might “suffer” more???

- [p. 21] “TensorFlow respectively the CIFAR-10 application and GROMACS” => “TensorFlow, the CIFAR-10 application in particular, and GROMACS”

- [p. 21] “make more advices what kind of datasets can have impacts on different tools” => “provide more suggestions on the types of datasets that can have impact on different tools”

Moreover, references [2], [3], [4], [24], and (mainly) [25] are missing important data for the reader to find the corresponding resources. Finally, there are some issues with some figures:

- [fig.1:] The “Container engine” box is highlighted with no reason.

- [fig.3,4,5:] The fonts are blurred, a better image quality is required.

- [fig. 3,4,5:] For which data set (small, medium, large) are these numbers?

**Have all data underlying the figures and results presented in the manuscript been provided?**

Reviewer #1: Yes

Reviewer #2: Yes

PLOS authors have the option to publish the peer review history of their article (what does this mean?). If published, this will include your full peer review and any attached files.

Reviewer #1: **Yes: **Ahmed BaniMustafa

Reviewer #2: No
---

## [Decision Letter · Decision Letter 1]

29 Apr 2021

Dear Dr. Krüger,

Thank you very much for submitting your manuscript "Performance and scaling behavior of bioinformatic applications in virtualization environments, to create awareness for the efficient use of compute resources." for consideration at PLOS Computational Biology. As with all papers reviewed by the journal, your manuscript was reviewed by members of the editorial board and by several independent reviewers. The reviewers appreciated the attention to an important topic. Based on the reviews, we are likely to accept this manuscript for publication, providing that you modify the manuscript according to the review recommendations.

Sincerely,

Christos A. Ouzounis

Associate Editor

PLOS Computational Biology

Edwin Wang

Benchmarking Editor

PLOS Computational Biology

[LINK]

Reviewer's Responses to Questions

**Comments to the Authors:**

Reviewer #1: Thank you for addressing my earlier comments. I have also noticed that you have included more tools and literature which is quite good.

However, I have noticed that the overhead numbers in table 6. (page 18) are quite different from those reported in the earlier submission! Can you explain? There is also a difference in the scoring of IDBA in table 7. (Page 17)! Can you explain?

Reviewer #2: The authors have done significant rewriting and performed extra experiments, alleviating many of the originally pointed issues. Now the categories of software tested are better represented, the figures are more informative (for example showing the corresponding error margins) and the text more compact, in general. However, I believe that the tools descriptions should be much shorter. I cannot understand why two (relatively large) paragraphs are needed for each of the tools. These are widely known software packages and the authors also provide references to the corresponding papers that contain all the details. In my opinion these descriptions could and should be much smaller (a small paragraph for each of the tools should be more than enough). Moreover, I would like to hear more details (at least some insights) about which characteristics of the datasets are the ones that create larger execution times (since the disk size is not the case). Finally, my concern about if this is the right venue, since the paper does not provide new biological insights (as described in the PLOS Comp. Biol. scope) remains. As I said, this is not crucial from my side (I have no strong opinion), however, if the authors could argue against this comment, their arguments could be of significant interest and it would be nice to highlighting the corresponding points in the manuscript.

**Have the authors made all data and (if applicable) computational code underlying the findings in their manuscript fully available?**

Reviewer #1: Yes

Reviewer #2: **No: **

PLOS authors have the option to publish the peer review history of their article (what does this mean?). If published, this will include your full peer review and any attached files.

Reviewer #1: **Yes: **Ahmed BaniMustafa

Reviewer #2: No

Figure Files:

Data Requirements:

Reproducibility:

References:

---

## [Decision Letter · Decision Letter 2]

2 Jul 2021

Dear Dr. Krüger,

We are pleased to inform you that your manuscript 'Performance and scaling behavior of bioinformatic applications in virtualization environments, to create awareness for the efficient use of compute resources.' has been provisionally accepted for publication in PLOS Computational Biology.

Best regards,

Christos A. Ouzounis

Associate Editor

PLOS Computational Biology

Edwin Wang

Benchmarking Editor

PLOS Computational Biology

Reviewer's Responses to Questions

**Comments to the Authors:**

Reviewer #1: No further comments.

Reviewer #2: Except one, all the issues raised by my side have been resolved, at least to an extent. The unresolved issue is about whether the paper is in the scope of the journal. I was asking for a rationale that indicates why this work provides "profound new biological insights", as PLOS Comp. Biology dictates for all Research Articles (https://journals.plos.org/ploscompbiol/s/journal-information#loc-scope). I was hoping that the authors could argue about this and that they would include a relevant discussion in their manuscript (e.g., in the excerpt where they summarise the contribution of their work). Instead of this, the authors provided an alternative interpretation of the text included in the scope webpage, which does not make sense to me. Based on the absence of supporting arguments for the contrary, my initial concern remains valid. I believe that this type of work fits better (and will attract more attention) in another venue more focused on informatics. However, based on the nature of this issue, I do not want to insist on this, and I will let the editor to take the final decision. If the editor feels that this work fits well in this journal, I have no further objections for its publication.

**Have the authors made all data and (if applicable) computational code underlying the findings in their manuscript fully available?**

Reviewer #1: Yes

Reviewer #2: Yes

PLOS authors have the option to publish the peer review history of their article (what does this mean?). If published, this will include your full peer review and any attached files.

Reviewer #1: **Yes: **Ahmed BaniMustafa

Reviewer #2: No

---

## [Editor Report · Acceptance letter]

16 Jul 2021

PCOMPBIOL-D-20-01988R2 

Performance and scaling behavior of bioinformatic applications in virtualization environments, to create awareness for the efficient use of compute resources.

Dear Dr Krüger,

I am pleased to inform you that your manuscript has been formally accepted for publication in PLOS Computational Biology. Your manuscript is now with our production department and you will be notified of the publication date in due course.

With kind regards,

Agota Szep
